# Use of an ETEC Proteome Microarray to Evaluate Cross-Reactivity of ETVAX^®^ Vaccine-Induced IgG Antibodies in Zambian Children

**DOI:** 10.3390/vaccines11050939

**Published:** 2023-05-04

**Authors:** Cynthia Mubanga, Michelo Simuyandi, Kapambwe Mwape, Kennedy Chibesa, Caroline Chisenga, Obvious Nchimunya Chilyabanyama, Arlo Randall, Xiaowu Liang, Richard H. Glashoff, Roma Chilengi

**Affiliations:** 1Enteric Disease and Vaccine Research Unit, Centre for Infectious Disease Research in Zambia, Lusaka P.O. Box 34681, Zambia; 2Division of Medical Microbiology, Department of Pathology, Stellenbosch University & National Health Laboratory Service, Tygerberg Hospital Francie van Zijl Drive, P.O. Box 241, Cape Town 8000, South Africa; 3Water and Health Research Center, Faculty of Health Sciences, University of Johannesburg, P.O. Box 17011, Doornfontein 2028, South Africa; 4Division of Virology, School of Pathology, Faculty of Health Sciences, University of the Free State, 205 Nelson Mandela, P.O. Box 339, Bloemfontein 9300, South Africa; 5Antigen Discovery Inc., 1 Technology Dr., Suite E309, Irvine, CA 92618, USA

**Keywords:** antibody cross-reactivity, microarray, ETEC, ETVAX^®^

## Abstract

Developing a broadly protective vaccine covering most ETEC variants has been elusive. The most clinically advanced candidate yet is an oral inactivated ETEC vaccine (ETVAX^®^). We report on the use of a proteome microarray for the assessment of cross-reactivity of anti-ETVAX^®^ IgG antibodies against over 4000 ETEC antigens and proteins. We evaluated 40 (pre-and post-vaccination) plasma samples from 20 Zambian children aged 10–23 months that participated in a phase 1 trial investigating the safety, tolerability, and immunogenicity of ETVAX^®^ adjuvanted with dmLT. Pre-vaccination samples revealed high IgG responses to a variety of ETEC proteins including classical ETEC antigens (CFs and LT) and non-classical antigens. Post-vaccination reactivity to CFA/I, CS3, CS6, and LTB was stronger than baseline among the vaccinated compared to the placebo group. Interestingly, we noted significantly high post-vaccination responses to three non-vaccine ETEC proteins: CS4, CS14, and PCF071 (*p* = 0.043, *p* = 0.028, and *p* = 0.00039, respectively), suggestive of cross-reactive responses to CFA/I. However, similar responses were observed in the placebo group, indicating the need for larger studies. We conclude that the ETEC microarray is a useful tool for investigating antibody responses to numerous antigens, especially because it may not be practicable to include all antigens in a single vaccine.

## 1. Introduction

Enterotoxigenic *E. coli* (ETEC) is a significant cause of moderate-to-severe diarrhoea (MSD) in children under five years of age in low-to-middle-income countries (LMICs) and is a leading cause of diarrhoea in travellers to endemic areas including tourists and military personnel [1,2,3,4] The highest incidence and mortality are reported in children below two years of age [1,3]. ETEC is estimated to be responsible for approximately 75 million episodes of diarrhoea in under-five years of age children annually, resulting in more than 18,700 deaths [1,5,6]. Data from a Zambian study looking at the aetiological agents of MSD in Zambian under-five years of age children confirm the significant contribution (40.7%) of ETEC to diarrhoeal disease [7].

ETEC is a Gram-negative, facultative aerobic, rod-shaped coliform of the genus *Escherichia* with strains comprising a phenotypically and genetically diverse pathotype [8,9]. In the classical paradigm of ETEC diarrhoea pathogenesis, the disease is caused by bacterial adherence to enterocytes using colonisation factors (CFs) or coli surface antigens (CS) and producing one or both of two plasmid-encoded enterotoxins: heat-labile (LT) or heat-stable (ST) [8,10,11]. The enterotoxins mediate the deregulation of the membrane ion channels in the epithelial cell membrane, leading to the loss of ions and large amounts of water with ST strains being responsible for most cases of MSD [8,12]. The CFs occur at various frequencies in different parts of the world and more than 25 have been identified, with the most common being CFA/I, CS1, CS2, CS3, CS4, CS5, CS6, CS7, CS14, CS17, and CS21 [8]. Enterotoxigenic *E. coli* fimbriae are divided into classes based on the phylogenetic relatedness of their major pilin subunits, with most of them belonging to either class 1 or class 5 (13). Class 1 includes CS12, CS18, and CS20, while class 5 includes CFA/I, CS1, CS2, CS4, CS14, CS17, CS19, and PCF071 [13,14]. Class 5 fimbriae are said to account for 30% of all clinical isolates [14,15]. The CFs are also classified into three families based on shared cross-reactive epitopes: CFA/I-like (CFA/I, CS1, CS2, CS4, CS14, and CS17), CS5-like (CS5, CS7, CS18, and CS20), and Class 1b (CS3, CS6, CS10, CS11, and CS12) [16,17,18,19].

Apart from the classical antigens (CFs and LT), other antigens have been suggested to be involved in infection [20,21,22], including two plasmid-encoded antigens, EatA and EtpA, that seem to be conserved across the ETEC pathovar [23].

Vaccines remain the most pragmatic way of ensuring ETEC prevention and control besides the provision of improved sanitation systems and clean water supply, which is not readily achieved in LMICs [24]. Treatment with antibiotics also has its challenges due to the growing problem of antibiotic resistance [3]. Several vaccines for ETEC are in development, and these range from subunit vaccines such as multiple epitope fusion antigens and ST toxoids, to whole-cell vaccines which include inactivated fimbriated ETEC, live attenuated ETEC expressing CFs, and *Shigella* vector expressing ETEC CFs [12].

A major hurdle in the vaccine development pathway is the antigenic diversity of ETEC strains. Hence, most candidate vaccines in development adopt a polyvalent approach targeting multiple CFs and LT (classical antigens) to achieve broad coverage [10,25].

ETVAX^®^, the most advanced candidate vaccine in clinical development contains monovalent bulks of *E. coli* strains developed using recombinant plasmids expressing the entire CFA/I, CS3, CS5, or CS6 operon and that have been inactivated either by mild formalin or mild phenol treatment [26,27]. The formulation of ETVAX^®^ includes LCTBA, a protein comprising the B subunit of the cholera toxin B (CTB) and the B subunit of the *E. coli* heat-labile toxin (LTB). Furthermore, the double mutant LT (dmLT) toxoid developed by John Clements serves as an adjuvant [28].

Along with other whole-cell vaccines in development, such as the ACE 527 and the *Shigella*-ETEC multivalent vaccines, ETVAX^®^ is designed to express some of the most prevalent and clinically relevant CFs with the view that these cover over half of all clinical isolates by inducing cross-reactive antibodies that may be able to prevent mucosal binding of both non-vaccine and vaccine-related CFs, thereby increasing protection against ETEC strains expressing CFs within the same family [18,23]. A study by Vidal et al. estimated that vaccines based on the major CFs (CFA/1, CS1, CS2, CS3, CS4, CS5, CS6) may prevent diarrhoea attributed to 66% of ETEC strains expressing ST alone and those expressing both ST and LT, strains that are largely responsible for clinical disease [12]. The ETVAX^®^ vaccine has been previously shown to induce cross-reactive antibodies to multiple CFs in the CFA/I and CS5 families [18,29] and has recently undergone safety and immunogenicity assessment in Zambia.

Protection against ETEC is understood to be mainly mediated by antibodies directed against the different CFs and LT produced locally at the gut mucosa, resulting in the development of protective immunity against homologous strains [11]. ST does not stimulate a strong immune response and is therefore considered poorly immunogenic even though it is implicated in most clinical diseases [11,25]. Studies of natural and experimental infections have revealed that the immune response to ETEC infection is more complex and more widely focused than previously appreciated [30], involving multiple antigens in addition to LT and CFs [31,32]. Both IgG and sIgA function as effector molecules of the mucosal immune system in the small intestine with sIgA being more abundant, while IgG levels increase with infection [33]. Systemic IgG and IgA have also been shown to be Important for protection, and it is believed that parenteral vaccine-induced serum IgG antibodies seep onto intestinal mucosa and prevent diarrhoea due to enteric bacteria [12,30].

The exploratory study reported here used the ETEC proteomic array technology to study IgG responses in the plasma of children participating in a randomised controlled phase 1 trial for ETVAX^®^ in Zambia to generate hypotheses for further studies. All the proteins overexpressed in ETVAX^®^, except for CS5, are included on the array as purified proteins. Our study evaluated whether ETVAX^®^-induced IgG antibodies are cross-reactive against ETEC antigens that are absent from the ETVAX^®^ vaccine.

## 2. Materials and Methods

This study utilised plasma samples collected from children aged 10–23 months participating in a single-site, double-blind, placebo-controlled, age-descending phase 1 clinical trial examining the safety, tolerability, and immunogenicity of an oral inactivated ETEC Vaccine (ETVAX^®^) adjuvanted with dmLT. By this age, it is expected that most maternal antibodies would have waned [34,35,36]. The distribution of trial participants to the different study arms is shown in the clinical trial flowchart adapted from the trial manuscript (unpublished) (Appendix A). The participants in this study were drawn from cohort B, which consisted of three groups of 20 participants each, receiving either ¼ dose of the adult vaccine dose, 1/8 dose, or a placebo. The adult dose of ETVAX constituted 150 mL of 1 × effervescent buffer, 80 × 10^9^ bacteria, and 1 mg of LCTBA administered orally with 10 g of dmLT. Furthermore, 2.5 g dmLT in 10 mL of effervescent buffer was used to administer the 1/8 and ¼ dose.

Each participant received three doses of the allocated intervention on the first day (D1), after two weeks (D15), and three months later (D90).

### 2.1. Sample Collection

Blood samples were collected at baseline prior to vaccination (D1 or V1), seven days after the second dose (D22 or V5), and seven days after the third dose (D97 or V7). The blood samples were centrifuged at 2000× *g* for 15 min, after which plasma samples were aliquoted and stored frozen at −80 °C before testing.

For this study, a total of 20 participants (4 from the placebo group and 16 from the vaccine group) were randomly selected by listing all cohort B sample IDs in an Excel spreadsheet and then grouped by vaccine allocation, i.e., ¼ dose of the adult vaccine dose, 1/8 dose, or a placebo. Every third sample in each column was then picked. A pre-vaccination (D1 or V1) and post-vaccination (D97 or V7) plasma sample from each of these was shipped to Antigen Discovery Incorporated (ADI) in the USA for analysis.

### 2.2. Lab Analysis

#### 2.2.1. Microarray Creation

The microarray was created at ADI as previously described [32]. Briefly, clones representing 4168 selected gene features encoding known ETEC antigens and surface proteins present in more than 40% of the ETEC isolates examined and not present in the genomes of three common *E. coli* commensal isolates were expressed in a cell-free in vitro transcription–translation (IVTT) system with each protein having a 5′ polyhistidine (HIS) epitope and 3′ hemagglutinin (HA) epitope. The IVTT proteins were then printed on nitrocellulose-coated glass slides using a robotic printer. The microarray and IVTT proteins were then validated accordingly before use.

#### 2.2.2. Sample Analysis

Test plasma samples along with control samples were added to the microarray and incubated. In addition to the study samples, additional arrays were probed following the same protocol with both negative and positive controls. The negative control was buffer only with no serum added. If any spots were reactive against the negative control, it indicated the cross-reactivity of the secondary antibody with the expressed protein. The positive control was a pool of reactive samples probed on each day of probing to assess the consistency across days. The antibody–antigen reaction was then detected and quantified by the GenePix^®^ 4300 Microarray Scanner (Molecular Devices, San Jose, CA, USA). All arrays were scanned and quantified, and then automated data extraction and QC was performed using R [37]. The raw signal for each spot was obtained by taking the raw intensity and subtracting the local background intensity. Normalized signals were obtained by first calculating the ratio of the raw spot signal to the sample-specific median of IVTT control spot signals, and then applying the base-2 log transformation. For purified recombinant proteins that do not have background signals from the IVTT system, raw signals were transformed by applying the base-2 log transformation.

From the raw data, automated QC metrics were calculated using R [37] for each array to identify arrays with unusual variation in control spots, spots with lower reactivity than IVTT control spot median, unusual background variation, and signal saturation.

#### 2.2.3. Data Analysis

IVTT-expressed protein data were separated into subsets by first identifying reactive antigens and filtering non-reactive spots from subsequent analysis (raw and normalized data for all spots were still retained). Reactivity filtering was performed by defining seropositivity as a normalized signal of 1.0 or greater. This corresponded to twice the sample-specific median IVTT control spot signal, i.e., background. Antigens were categorized as reactive and carried forward for statistical analysis if at least 1 sample was seropositive among any participant dose group. It is worth noting that raw and normalized data were still retained and published in a repository for all array spots. For purified protein, the normalization was only applying the base-2 log transformation. No reactivity sub-setting was performed on the purified proteins and all 39 spots were carried forward for statistical analysis.

#### 2.2.4. Statistical Analysis

Paired *t*-tests of visit 1 (pre-vaccination) and visit 7 (post-vaccination) samples were performed to assess the vaccination effect in each treatment group separately using R statistical software [37]. Independent *t*-tests comparing the means of different treatment groups at specific visits, as well as the increases (deltas) from visits 1 to 7, were used to assess differences between groups. The *p*-values reported in the abstract, text, and figures are all based on paired or independent *t*-test. Appendix A contain additional results including the group means, area under the receiver operating characteristics (ROC) curve (AUC), and *p*-value of non-parametric Wilcox ranks testing. All reported *p*-values are raw and are not corrected for multiple testing [38].

## 3. Results

Twenty randomly selected participants aged between 10 and 23 months were included in this study. Of these, 16 received the ETVAX^®^ vaccine (a vaccine consisting of four inactivated *E. coli* bacterial strains over-expressing the colonisation factors CFA/I, CS3, CS5, and CS6, respectively, the toxoid LCTBA, and dmLT serving as an adjuvant) and 4 received the placebo. A total of eight vaccinees received one-quarter of a dose while the other eight received one-eighth of an adult dose. Appendix A gives more information on the characteristics of the study participants.

### 3.1. Responses to Purified Proteins

#### Microarray Responses to ETVAX^®^ Antigens and Non-ETVAX^®^ Antigens

The ETEC microarray contained purified proteins including full native fimbriae, as well as major and minor colonization factor antigen (CFA) domains, putative colonisation factors, and fimbrial surface antigens as listed in Appendix A [39] Notably absent from the array is CS5 and members of the CS5-like family [17,18,19]; however, all other proteins overexpressed in ETVAX are present.

Figure 1 shows the IgG responses to all purified proteins present on the array with the class 5 fimbriae ordered by subclass and the remaining proteins ordered alphabetically. EtpA (exoprotein adhesin), YghJ (metalloprotease), and CS3 (native fimbriae) had the highest average signal intensities considering the pre- and post-vaccination samples across all subjects. Nearly all subjects have very high antibody levels against EtpA (N-terminal) and YghJ, while the pre-vaccination antibody levels against CS3 exhibit much more variation. Similarly, pre-vaccination antibody levels against other CFs vary widely across subjects: around half of the subjects have very high levels against CFA/I, and less frequent high levels are observed for CS17, CS19, CS2, CS14, and CS6.

Figure 2 presents the results of three different pairwise comparisons in panels A, B, and C with the top 10 purified protein spots ranked by absolute mean difference. The identifier for each protein is followed by the *p*-value. The vaccinated individuals generally had stronger IgG responses post-vaccination (V7) compared to pre-vaccination (V1) with all but CS6 having V7responses that were significantly higher than V1 (*p* < 0.05) (Figure 2A). The amount of increase varies widely across individual subjects. In general, the largest increases are observed for subjects starting from very low pre-vaccine levels and subtle increases are observed when pre-vaccine levels are already high. This is seen most clearly in row 2 of the heatmap (Figure 1) focusing on the CFA/I responses in the one-quarter of a dose group where the three rightmost subjects have clearly the lowest starting antibody levels (left panel) and the increases for these three subjects are from 16 to 32-fold (rightmost panel), whereas the other five subjects are very high pre-vaccination and have increases of four-fold or less. See also Appendix A, which contains the statistics for these comparisons. Vaccine antigens CFA/I, CS3, CS6, and LTB, and non-vaccine antigens CS4, CS14, and PCF071 were the top 10 proteins in the vaccine group. The presence of strong IgG-increased responses to CS4, CS14, and PCF071 in the vaccine group is suggestive of vaccine-induced cross-reactive antibodies to these proteins.

In the placebo group (Figure 2B), the top 10 purified proteins included CfaEad (CFA/I adhesin domain truncate), CS1, CS3, CS4, CS14, CS17, CS19, LTA, and CsbDad (CS17: adhesin domain truncate), of which only CFA/I and CS3 are highly expressed by the vaccine bacteria. LTA is also covered in the dmLT administered in the vaccine. V7 responses were higher than V1 for all antigens except CfaEad; however, unlike with the vaccine group, none of these differences were statistically significant, and the higher averages are driven by a single high outlier (S087).

Figure 2C shows the comparison of purified protein antibody mean deltas (increase from V1 to V7) difference among all placebo and all vaccinated individuals for the top 10 antigens. Generally, the largest delta mean differences can be observed in the vaccinated compared to the placebo group with vaccine antigen responses tending to have more positive delta values (see Appendix A) with the exception of CS19 and CS17 for which the placebo group had positive delta changes.

The error bars for CFA/1, CS6, and CS3 major CstG are not overlapping, with *p* values of less than 0.05, suggesting the possibility that these post-vaccination responses could be attributed to the vaccine.

### 3.2. Microarray Responses to Other ETEC Proteins

The microarray included other proteins in addition to the fimbriae-purified proteins listed in Appendix A. These comprised cell surface proteins, enzymes, structural proteins, and transmembrane proteins, among others. The heatmaps in Figure 3 show the changes in IgG antibody intensities to selected IVTT proteins that have previously been reported to show increases in ALS IgA following a challenge with ETEC H10407 [10,32]. We observed that the IgG responses to some of these proteins appeared to decrease substantially between V1 and V7. We attribute this to the possible waning of maternal antibodies that could have been detected at baseline.

## 4. Discussion

The development of a broadly protective ETEC vaccine has been a challenge due to the huge diversity in ETEC pathovars. This study thus reports an evaluation on the ability of ETVAX^®^, the most clinically advanced ETEC vaccine candidate, to induce cross-reactive IgG antibodies against non-vaccine antigens.

Firstly, the results from our study affirm the high and early exposure to ETEC and possibly other pathogenic *E. coli* among Zambian children. This can be seen in the high intensities of IgG antibodies to many ETEC antigens and proteins at baseline (pre-vaccination) in the vaccine group and pre-and post-vaccination time points in the placebo group (Figure 1). ETEC has been previously reported to be one of the earliest symptomatic enteric illnesses among children in endemic areas [1,40]. While we acknowledge that the observed IgG responses may be due to the presence of maternal antibodies, various studies have shown that maternal antibodies rapidly wane beyond the age of six months, and therefore safely assume that the antibodies detected in this study, particularly at V7, are likely due to the children’s natural exposure to ETEC [34,35,36,41]. In Appendix A, we do see that a few of the participants did have a positive ETEC colony PCR test during the study period, and, in Figure 2B, we observe general increases in IgG to the 10 most reactive proteins in the placebo group. These results therefore show the need to vaccinate children younger than 6 months as they may benefit more from the vaccine, as this may provide early protection from infections.

The immunodominant purified proteins with very high IgG responses included EtpA, EatA, YghJ, CFA/I, CS1, CS2, CS3, CS4, CS6, CS14, CS17, CS19, PCF071, LTA, and LTB. The CFs here are compared to those observed in a previous Zambian study [17]. Most of the immunodominant CFs are members of class 5 fimbriae (both major and minor subunits) which are involved in the adherence of the ETEC bacteria to the enterocytes [18]. Unlike the rod-like structure of class 5 fimbriae, CS3 are fine flexible fibrillae, while CS6 is afimbriae and featureless [12,42,43]. These CFs are common and occur in many clinical isolates [12].

We also see that the A subunit of LT is immunodominant (Figure 1 and Figure 2B) despite some potential vaccines only including the B subunit of the LT toxin (LTB), which is presumed to be more immunogenic in their formulations [44]. A study by Norton et al. demonstrated that the two subunits are immunogenic and acted synergistically in neutralizing the LT toxin action [44]. Therefore, the addition of dmLT which contains LTA to ETVAX^®^ enhances its coverage [28,45,46].

EatA, a serine protease autotransporter involved in mucin degradation enabling bacteria access to the epithelium [23]; EtpA, a secreted exoprotein adhesin that functions as a molecular bridge between the bacterial surface and its appendages; and YghJ, a metalloprotease that digests intestinal mucin, are all highly immunodominant [23]. These proteins seem to be conserved across various ETEC strains and other pathogenic *E. coli* and have been reported to be involved in ETEC pathogenesis with the potential for inclusion in vaccines [23,32,47,48,49].

We also observed that three non-vaccine CFs (CS4, CS14, and PCF071) which belong to class 5 fimbriae were among the top ten reactive antigens in the vaccinated group. Post-vaccination responses were statistically significantly higher than pre-vaccination for all three with *p*-values of less than 0.05 (*p* = 0.0039 for PCF071, *p* = 0.028 for CS14, and *p* = 0.043 for CS4). While CS4 and CS14 were also among the top ten antigens in the placebo group, we believe that the responses observed in the vaccine group may have been vaccine-induced as CS4 and CS14 belong to the same class of fimbriae as CFA/I, making the cross-reactivity of antibodies highly likely. However, it is difficult to conclude given the small study sample size.

A similar study by Leach et al. reported that the cross-reactivity of ETVAX^®^ derived antibodies against non-vaccine CFs, namely CS1, CS14, CS17, and CS7, of which CS1, CS14, and CS17 are members of class 5 fimbriae [14,18]. Another similar study by Svennerholm et al. also reported the cross-reactivity to CFA/I family CFs (CS1, CS14, and CS17) in faeces among CFA/I responders, and they also observed high responses to CS7 among CS5 responders [29]. In our case, CS17 and CS1 did not make the top 10 antigens in the vaccine group, likely due to the small sample size, and therefore possible cross-reactivity could not be detected.

The two studies above differ from our study in that they had larger sample sizes and mucosal IgA responses were measured using ELISA; while, in our case, plasma/systemic IgG responses were measured using a protein microarray. Nonetheless, in all three studies, the cross-reactivity to non-vaccine antigens belonging to the same class of fimbriae or family can be seen.

The use of the microarray in this study allowed us to efficiently screen a range of proteins and identify other ETEC proteins that are highly immunogenic and possibly play a role in ETEC pathogenesis apart from the classical antigens. Among them were putative membrane proteins, conserved hypothetical proteins, putative transmembrane proteins, putative antigen43 precursor, adhesin autotransporter, peptidoglycan-associated lipoprotein, and putative flagellin (Flic H11). While these proteins, particularly EtpA, EatA, and YghJ, exhibit some protection in preclinical/animal studies, their role as protective antigens in human infection is yet to be clearly shown [6,32]. We also observed drops in the IgG to fliC, antigen 43, and some other IVTT proteins (Figure 3) among several participants, possibly due to the waning of maternal antibodies that may have been present before vaccination [34,36,41,50].

Our study, though exploratory, provides an extensive look at the IgG response to ETEC infection (pre-vaccination/placebo) and vaccination, and it is the first study from a disease-endemic area to report the use of the ETEC microarray. However, our study had several limitations. Firstly, due to funding constraints, we had a very small sample size and therefore the results of this study are inconclusive and may not be generalizable. Secondly, the microarray used did not include CFs from the CS5-like family, i.e., CS5, CS7, CS18, and CS20, and therefore possible cross-reactivity from CS5 could not be assessed. Another limitation was the lack of assessment of secretory IgA responses which are regarded as paramount in ETEC infection. However, we believe the IgG responses do still enable us to answer our main question of whether ETVAX^®^ induces a cross-reactive response. In addition, we observe that plasma IgG and IgA responses to LTB measured by ELISA in the clinical trial were significantly higher in the vaccine group compared to the placebo group as observed in this study (Appendix A, Appendix A). Given the long duration between pre-and post-vaccination assessments, it is difficult to rule out the influence of natural infection responses on the observed post-vaccination responses. Our post-vaccination sample was collected 7 days after the third dose and this may have been too soon to accurately measure a vaccine-induced IgG response in view of ongoing background infections.

For future studies, we would recommend a larger sample size with substantial numbers of participants in both the vaccine and placebo groups to be able to come up with an effective evaluation of vaccine-induced response. We would also recommend that studies of this nature conducted in an endemic area include in their study design systems such as having substantial numbers in the control group to control for background responses that may be due to natural infection.

## 5. Conclusions

In conclusion, we have shown that the ETEC microarray is a tool that can be used to study/analyse antibody responses to numerous antigens from complex microorganisms such as ETEC. We observed that ETVAX^®^ has potential to induce IgG antibodies that are cross-reactive against some ETEC antigens that are not overexpressed in the vaccine, even though our results are not conclusive due to the small sample size and call for the need for a bigger study [18].

Not much is known about the ETEC strains that occur in our setting. However, this study provides information about the plasmid-encoded proteins such as CFs, EtpA, and EatA that occur in the strains in Zambia, thereby providing information for vaccine developers.

We have also shown that various other proteins are involved in ETEC pathogenesis and may be investigated for their role in protection.

## Figures and Tables

**Figure 1 vaccines-11-00939-f001:**
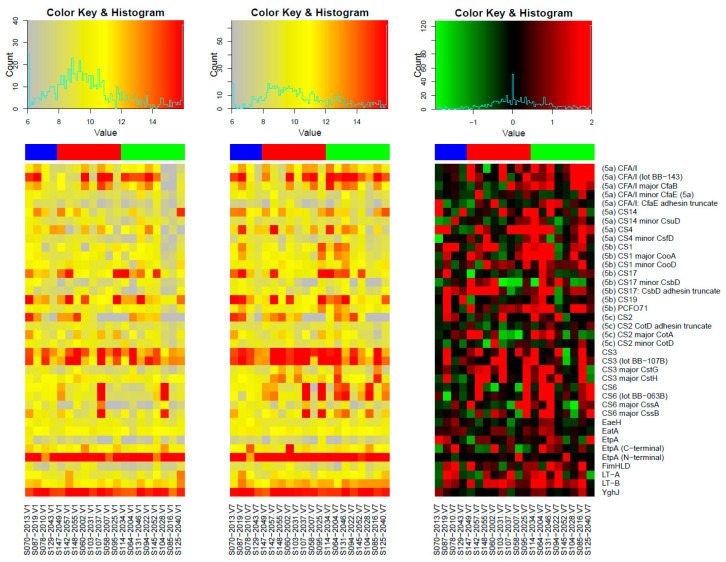
The heatmaps present the base-2 logarithm of the individual reactivity levels at Day 0 in the left panel and Day 97 in the centre panel. The delta values (change from Day 0 to Day 97) are shown in the right panel. The heatmap colouring of the “delta” panel highlights both decreases (green) and increases (red) within a range of [−2, 2]. Values outside the range are set to the range limit for the heatmap. The class 5 fimbriae are ordered first by subclass, and then proteins are ordered alphabetically. The subjects are ordered by group where blue corresponds to placebo, red to 1/8 dose, and green to 1/4 dose.

**Figure 2 vaccines-11-00939-f002:**
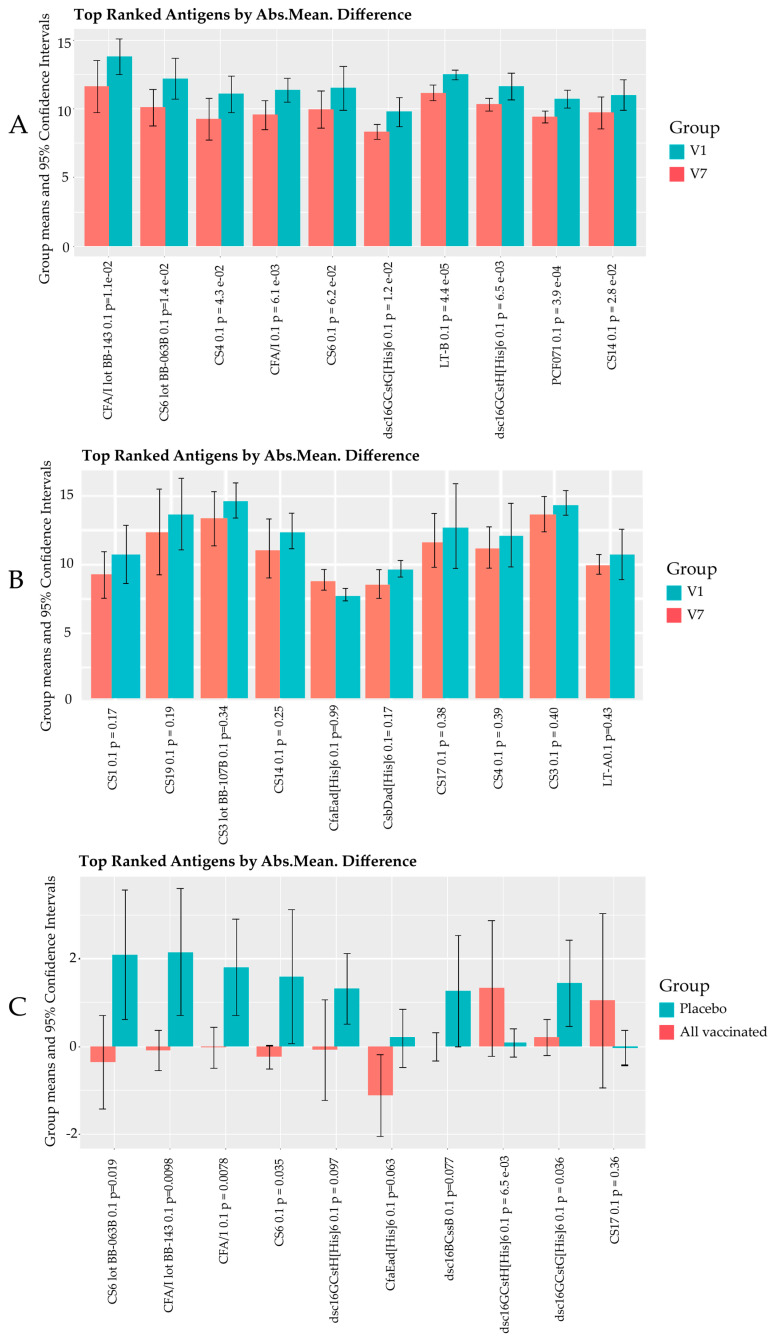
Graphs of IgG responses to the top 10 purified antigens/proteins in the microarray. (**A**) Comparison of the averages of all vaccinated individuals at visit 1 and visit 7 for each antigen. (**B**) Comparison of the averages of the placebo group at visit 1 and visit 7 for each antigen. (**C**) Comparison of the delta changes in averages of all vaccinated individuals compared to the placebo group at visit 1 and visit 7 for each antigen. The vertical error bars define the 95% confidence interval around each average (i.e., there is a 95% probability that the true unknown mean lies somewhere in the defined interval based on the sample average and standard deviation).

**Figure 3 vaccines-11-00939-f003:**
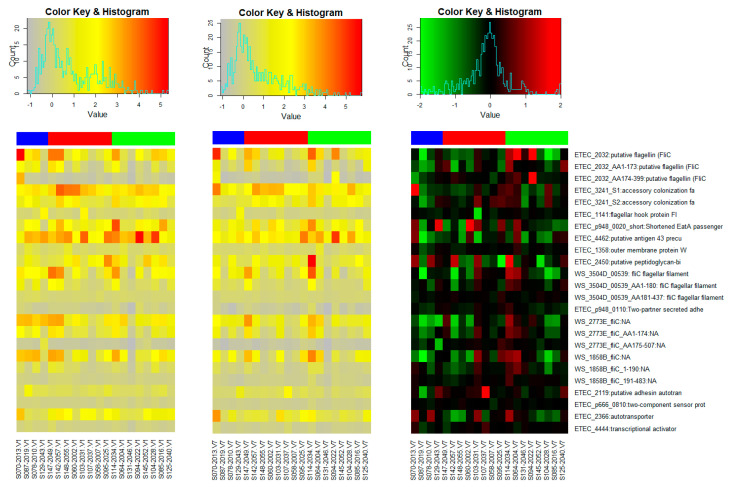
The heatmaps present the normalized intensity base-2 logarithm of the individual reactivity levels at Day 0 in the left panel and Day 97 in the centre panel. The delta values (change from Day 0 to Day 97) are shown in the right panel. The heatmap colouring of the “delta” panel highlights both decreases (green) and increases (red) within a range of [−2, 2]. Values outside the range are set to the range limit for the heatmap. The subjects are ordered by group where blue corresponds to placebo, red to 1/8 dose, and green to ¼ dose.

## Data Availability

The data are available in the NCBI Gene Expression Omnibus (GEO) repository. The GEO series accession number is GSE220814.

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
