# Peer review of "Use of an ETEC Proteome Microarray to Evaluate Cross-Reactivity of ETVAX® Vaccine-Induced IgG Antibodies in Zambian Children"

_vaccines, 2023, doi:10.3390/vaccines11050939_

Round 1

Reviewer 1 Report

 Review of:  Cynthia Mubanga et al.  Use of an ETEC proteome microarray to evaluate 2 cross-reactivity of ETVAX® vaccine-induced IgG antibodies in Zambian children.

General

Mubanga et al have examined plasma samples from Zambian 16 infants 12-23 months old, before vaccination and 7 days after the third dose of ETVAX vaccine, thus 97 days after the first dose. 4 infants given placebo acted as controls. IgG antibody responses were analysed in The US using a proteome microarray panel containing all major CF, LT and other ETEC proteins, except CS5.

Major findings are consistently high baseline antibody levels against EtpA, YghJ and CS3 in these 20 infants, augmented post-vaccination antibody responses against CFA/I, CS3, CS6, and LTB among the 16 vaccinated infants.  Their also mention finding significantly higher post-vaccination responses CS4, CS14, and PCF071 among vaccinated infants.

As has become increasingly clear, there are high background levels to many ETEC antigens in infants in endemic areas. Combined with a low number of participants, and maybe sampling timepoints, the results are somewhat difficult to interpret.

The manuscript is well written and figures illustrative. Information regarding antibody profile in Zambian children of at this age is interesting and informative. Especially, the study shed light on responses against less well-known ETEC antigen in an ETEC endemic setting.

Abstract

Concise and good, but I do not think there is good reason to conclude that ETVAX induces cross-reactive IgG antibody responses to CD4, CD14. Possibly it can be true for PCF071, which did not turn up among the top10 differential responses the placebo group, but considering this was only 4 individuals it is also a rather bold statement. I think the authors need to be far more cautious about the interpretation of that finding, and it is not suited to form the main conclusion of the paper.

Introduction

The introduction is a bit lengthy but summarizes well many relevant aspects of ETEC epidemiology and antigenicity. It is well written, cites relevant literature, generally good English language.

Methods

Were lab personnel blinded regarding the vaccination status and timepoints?

Line 165: unclear sentence “…spot foreground less median local background..”. Please explain this better for readers who are not familiar with the proteome array analysis.

Line 194: Authors state that “.. raw and false-discovery corrected p-values..”. I assume this means a form of multiple comparisons correction. The authors need to be more specific about what tests and types of corrections were made.

Results

Authors find significantly higher post-vaccination responses CS4, CS14, and PCF071 among vaccinated infants. However, considering similar, or slightly higher levels, as well as a non-significant increase from baseline, in the 4 placebo controls of CS4 and CS14 (based on figure 2B) it is problematic to suggest that the vaccine has cross-protective potential against these antigens.

Discussion

In general the discussion is well written and the authors do mention important limitations like the low number of infants, especially the placebo control group, and background exposure to ETEC in both groups affecting the results. However, they do not fully carry this caution into their interpretation, which should be more cautions especially with regard to the cross-protection of ETVAX against the CD4 and CS14. The statement in line 335 “… we still believe…” is not well founded. I think the results presented here are not supporting cross-protection. Such a conclusion could only be considered if the authors had found a significant difference, after multiple comparison correction, between the placebo and vaccinated group. With the data included here, I think vaccine cross-protection may be possible, but the study cannot tell us.

As the authors state, ETEC is clinically important from the early weaning period. At that age the infants may still have some maternal antibodies, but would perhaps benefit more from the vaccine than the age group included in this study, as they may not have encountered as many ETEC infections already. The relevance of the findings for the final target age group (infants at weaning) should be discussed.

The authors should also discuss the impact that the sampling timepoint may have had on their results. Is day 7 after the last vaccination a good timepoint to examine IgG responses?  At day 7 their results are more likely to be influenced by the general background infections occurring in the 2.5 months since the 2nd vaccination. I think they might have seen a more vaccine specific response had they sampled 14 or more days after the last vaccination.

Line 312: Fimbrial classification is a repetition form the introduction, and should be deleted in one of these sections.

Miscellaneous

Presenting p-values in the form p=2.8e–02, is both unusual, and needs more characters than writing the value plainly, with 3 decimals. I think this is unnecessarily confusing and should be changed to plain numbers, at least in the text.

Affiliation number 6 is not given.

Author Response

Thank you for the insightful comments.

Please see responses highlighted in yellow

Reviewer 2 Report

The manuscript (vaccines-2257989) entitled “Use of an ETEC proteome microarray to evaluate cross-reactivity of ETVAX vaccine-induced IgG antibodies in Zambian children” by Mubanga and colleagues describes the use of a protein microarray to detect antibody responses to a broad array of CFs and ETEC antigens in serum samples from children vaccinated with the ETEC vaccine ETVAX.  ETVAX is the most advanced ETEC vaccine in terms of clinical trials.  This study utilizes valuable samples from children in an endemic country to evaluate vaccine-induced responses and potential cross reactivity to additional non-vaccine antigens.  This is an important study with the potential to communicate significant data about the ETVAX vaccine in an endemic population.  Expanded presentation of the data is required to support the conclusions presented, especially with respect to induction of cross reactive responses.  The following comments are provided for consideration.

1.       Please speculate why the responses to full length EtpA were so much less than the N-terminal fragment.

2.       Figure 2 requires format modification to increase legibility – it is impossible to read as is, especially p values.  The labels under each bar set are too light.

3.       Throughout the manuscript and figures, p values should be shown conventionally as numbers with digits rather than with “e” format.

4.       Figure 2A and Supp Table 3 indicate that there is a significant difference in V1 vs V7 for all antigens shown.  It is not clear why CFA/I, LTB, CS3 and PCF071 are pointed out (line 235). These antigens are not necessarily the ones with the biggest difference between V1 and V7.  It would be more helpful if all like antigens (e.g. CFA/I lots) were grouped next to each other in the figure and in the table.  Supp Table 3 does not explain what “increase to 50Pct Count” means or how this is relevant.

5.       Most Important: Line 246 mentions “frequent” responses. While this data is not included (and this word should be deleted from this sentence) it would be very useful to include the number of >2 or >4 fold responders between V1 and V7 for each group.  This would be much more convincing than the data alone as shown.  This kind of data is indicated in Supp Table 5, but this seems to be ELISA data – although this is not indicated.  Without actually showing the data for frequency of responses, the conclusions about cross reacting responses to CS4, CS14 and PCF071 in vaccinees vs placebos cannot be made (lines 246-248).  Discussion Lines 334-337 would benefit from this data as well.

6.       Figure 2 would benefit from having the antigen bars in each panel (A, B, and C) be in the same order where the same antigens are present.  2C requires clarification of the Y axis.

7.       Is line 264 meant to read “purified antibody”?

8.       Line 278 should read strain “H10407”

9.       The Discussion mentions the ETEC positive individuals (Supp Table 1) during this study. Do the antibody reactivity responses correlate with these potential infections?

10.   Discussion Lines 334-337 would benefit from frequency data (see point 5) to support the conclusion about cross reactivity with non-vaccine CFs.  Frequency data on CS17 and CS1 would be useful as these antigens had previously shown reactivity (lines343).

11.   Line 360 refers to waning of maternal antibody which the authors argue strongly against in previous discussion.  This should be consistent throughout.

Author Response

Thank you very much for the very helpful and insightful comments. We did our best to address them as best as we could in the time given.

Please see attached document with responses highlighted yellow

Reviewer 3 Report

The paper described the evaluation of production of antibodies against ETVAX , the most advanced vaccine for Enterotoxigenic E. coli  that is under clinical trial.  They checked the safety, tolerability, and immunogenicity using a proteome microarray where IgG reactivity to over 4000 ETEC antigens and proteins were assessed. They found  high IgG responses to a variety of ETEC proteins including classical ETEC antigens (CFs and LT) and non-classical antigens. They also found reactivity to CFA/I, CS3, CS6, and LTB to be stronger in the vaccinated subjects as compared to the placebo.

The paper is well written and the data has been presented very well using heat maps and histograms. This is the first time that the production of IgG against the vaccine ETVAX has been evaluated using microarray.   However, this can be published as a preliminary investigation since the number of subjects and samples for testing were rather limited.  The microarray used did not carry one of the major determinants for ETEC pathology, the CS5 fimbriae.  It was not clear how safety and tolerability of the vaccine was tested.  The authors only tested 2 doses, ¼ and 1/8th of the adult dose.  The tolerability of the vaccine was not really evaluated over a time.  The conclusions were drawn 90 days after vaccination. The maternal immunity also played a role in the evaluation as the weaning time was not considered as the subjects were infants.

Author Response

Thank you very much for your helpful and encouraging comments. 

Please see attached document with responses in yellow.

Round 2

Reviewer 1 Report

Authors have replied well and made important changes to most comments, except two issues that needs to be better addressed, and one new sentence that needs reformulation.

1. Authors have changed their abstract conclusion from

 "We conclude that ETVAX induces cross-reactive IgG antibody responses to non-vaccine CFs CS4, CS14, and PCF071 from the class 5 fimbriae and could provide a level of IgG antibody coverage beyond the core vaccine antigens themselves."

 to

 "We observe that ETVAX® has the potential to induce cross-reactive IgG antibody responses to non-vaccine CFs CS4, CS14, and PCF071 from the class 5 fimbriae and show that the ETEC microarray is a good tool for assessment of antibody responses to numerous antigens."

even if they themselves now write in the manuscript conclusion that  ”.... even though our results are not conclusive....”

I think it is unwise to put the inconclusive and speculative interpretation about cross reactive antibodies as a conclusion in the abstract. I do see that this would have been an interesting point to make for the authors, and maybe a hoped for result, but the obvious statement based on the data and the limitations would be to say something along the lines of f.ex ”We observed interesting increases in reactivity against non-vaccine CFs CS4, CS14, and PCF071 in the vaccinated group, but similar increases were also found in the small placebo group, so larger studies are needed to assess this issue”.

Rather than weakening their credibility with overselling this weak point, the study could focus on the application of the microarray to map responses against the main ETEC antigens which is already an important and interesting outcome, and that ETVAX was able to give higher reactivity to CFA/I, CS3, CS6, and LTB despite ETEC circulating in the community, causing considerable background reactivity also against the vaccine antigens.

2. The authors have added in line 204 onwards ”In addition, two statistical tests were performed on the normalized signals for each individual antigen: (1) t-test and (2) non-parametric Wilcox ranks test. Both the uncorrected p-values and p-values after controlling for false discovery due to multiple testing (39) are reported for both tests”.

I can find the t-test and wilcoxon p-values in tables S3 and S4 from revision 1, but I can not see anywhere that authors have given information about whether these were corrected or uncorrected for false discovery. There are no revised table S3 and S4 in the new submission. I therefore have a number of comments and questions for the authors:

1.       Authors need to clearly state which p-values have been false discovery corrected, and which have not.

2.       It is good to be transparent and give p-values for two types of tests, but authors should state which test did they chose to base their interpretation on, and have used in the mansucript?

3.       Authors have just given a 1995 reference about false discovery rate. It would be very helpful if they in the manuscript state the type/short name of the false discovery testing they apply, so the reader need not look up the paper to find out.

4.       The description of the statistics is done in two paragraphs. The first, starting at line 190 mentions the paired t-test and then laterr seems come back to this in more detail in line 205. Unless I have misunderstood this I think the statistical analysis paragraph can be made clearer by collecting  the information about the paired tests together.

5.       In the first paragraph authors also mention the independent t-test for comparing groups in line 192. Were any multiple comparison testing applied to these group comparisons?

6.       With groups of size 16 and 4 infants, it is difficult to assess normal distribution of data and many scientists would choose a non-parametric test like MannWhitney here. Were non-parametric test considered for the group comparisons as they were for the longitudinal comparisons?

3. The new sentence added in this revision, line315

”These results therefore show the need to vaccinate children younger than 6 months as they may benefit more from the vaccine, as it would provide a booster effect on the maternal antibodies”

It remains controversial and probably vaccine specific whether a vaccine response is affected by maternal antibodies, or inhibited by it (example pertussis). The formulation that the vaccine would provide a booster effect on the maternal antibodies is a bit unprecise. I agree that children younger than 6 months would benefit from the vaccine and that this oral vaccine can be speculated to elicit antibody production in infants, because natural infection seems to elicit infant antibody responses uninhibited by maternal antibodies. I suggest authors reformulate the sentence so it does not seem that the vaccine provides a booster effect on maternal antibodies, but rather that it may elicit infant antibodies before maternal antibodies wane.

Author Response

Thank you for the insightful and very valuable comments.

Please see our responses highlighted yellow in the attached document
